# Lipid Parameters and the Development of Chronic Kidney Disease: A Prospective Cohort Study in Middle-Aged and Elderly Chinese Individuals

**DOI:** 10.3390/nu15010112

**Published:** 2022-12-26

**Authors:** Shumei Liao, Diaozhu Lin, Qiling Feng, Feng Li, Yiqin Qi, Wanting Feng, Chuan Yang, Li Yan, Meng Ren, Kan Sun

**Affiliations:** Department of Endocrinology, Sun Yat-Sen Memorial Hospital, Sun Yat-Sen University, 107 Yanjiang West Road, Guangzhou 510120, China

**Keywords:** dyslipidemia, chronic kidney disease, urinary albumin-to-creatinine ratio, creatinine, estimated glomerular filtration rate, TG to HDL-C ratio

## Abstract

Epidemiological evidence suggests that lipid parameters are related to the progression of chronic kidney disease (CKD). Nevertheless, prospective studies that comprehensively assess the effect of routinely available lipid measures on the development of CKD are lacking. The aim of this study was to longitudinally assess the influence of lipid metabolism indicators on the presence of CKD in a large community-based population. We conducted a prospective cohort study at Sun Yat-sen Memorial Hospital, China, with 5345 patients of 40 years or older. Cox regression models were conducted, and hazard ratios (HRs) and 95% confidence intervals (CIs) were calculated to assess lipid parameters and their relationship with the incidence of CKD. During the follow-up period, 340 (6.4%) subjects developed CKD. The incidence of CKD increased progressively with quartile values of triglyceride (TG), the ratio of non-high-density lipoprotein cholesterol to high-density lipoprotein cholesterol (non-HDL-C/HDL-C) and the ratio of TG to HDL-C, but decreased with HDL-C quartiles (*p* < 0.0001 for all trends). Pearson’s correlation analysis and multiple regression analyses indicated that these parameters were also associated with various indicators of kidney function. Moreover, we found that among all the lipid parameters, TG/HDL-C emerged as the most effective predictor of CKD. In conclusion, our findings suggest that TG/HDL-C better predicts the incidence of CKD in middle-aged and elderly Chinese individuals than other lipid parameters tested in the study.

## 1. Introduction

In recent years, chronic kidney disease (CKD) has led to a significant increase in global mortality and an increased socioeconomic burden, making early screening and prevention of CKD particularly important [1]. CKD includes a range of pathophysiological alterations, such as inflammation, oxidative stress, endothelial dysfunction, uremia and hyperlipidemia, accompanied by the extensive and accelerated formation of arterial atherosclerotic plaque formation [2]. The detection of blood and urine indicators caused by these alterations can serve as a screen and predictor of CKD to some extent. Risk assessment criteria for kidney disease progression incorporate several pathophysiological alterations, including estimated glomerular filtration rate (eGFR) and proteinuria/albuminuria, but intraindividual variability in these indicators is high and sometimes does not imply true biological changes. Therefore, more sensitive and specific biomarkers need to be found to better predict CKD [3].

Dyslipidemia is common in patients with CKD. Patients with early-stage CKD are usually characterized by low levels of HDL-C, high levels of TG and normal or increased levels of low-density lipoprotein cholesterol (LDL-C); the availability of these markers may therefore increase the degree of accuracy of early screening for CKD [4,5]. We have previously done cross-sectional studies on the efficacy of lipid profiles with renal function, which suggested that there are discordant associations of lipid parameters with both albuminuria and renal damage [6]. In a web-based analysis, the authors identified the presence of multiple higher baseline levels of lipids in CKD patients, including glycerophospholipids, glycerolipids and sphingolipids, and suggested that disturbed lipid metabolism is an alteration that occurs before the onset of CKD [7]. A two-sample Mendelian randomization analysis in the largest lipid and CKD cohort revealed a positive causal relationship between hereditary HDL-C concentrations and renal function. Conversely, the relationship between hereditary altered LDL-C or TG concentrations and renal function was absent [8]. Other studies concluded that high TG levels, but not other lipid markers, were associated with albuminuria in the general Chinese population. The fact that TG and TG/HDL-C were negatively associated with eGFR and that TG was a better predictor of CKD in men was also emphasized [9,10]. Lipid parameters themselves have a relatively powerful predictive function. However, the importance of lipoprotein metabolism-related markers for predicting CKD remains inconclusive. With the growth of geriatric population, it is of great public health importance to identify subjects at greatest risk of developing early renal damage and to study the relationship of dyslipidemia with its occurrence and development. 

After a systematic review of such literature, we found a lack of relevant longitudinal studies that simultaneously explored the association between routinely available lipid measures and risk of CKD. Therefore, we conducted this study in a large community-based cohort in middle-aged and elderly individuals with the aim of longitudinally assessing the relationship of lipid markers with both urinary albumin excretion and renal function.

## 2. Methods

### 2.1. Population and Study Design

This cohort study was conducted in several communities in Guangzhou, China from 2011 to 2014–2016. The study’s subjects were obtained from a multicenter prospective observational study that aimed to assess chronic diseases in the Chinese population [11,12]. In all, 10,104 participants were enrolled who were 40 years or older and were recruited to attend through test notification or house visits. A total of 9916 subjects completed the consent form and were accepted into enrollment for the survey, representing a 98.1% participatory rate. While a total of 2917 were lost, 6999 were followed up with (including 125 deaths, 995 who could not attend the site but completed a brief questionnaire and 5879 who are currently in the follow-up database), resulting in a 71% follow-up rate. Participants from whom we were unable to collect data on HDL-C, LDL-C, and TG (n = 16) or information on baseline urinary albumin-to-creatinine ratio (UACR), high-grade albuminuria, CKD diagnosis (n = 466), or follow-up UACR and creatinine (n = 52) were excluded from the analysis. Eventually, 5345 patients met the criteria to stay in the final analysis (Figure 1). This research protocol was endorsed by the Institutional Review Board of Sun Yat-sen Memorial Hospital, Sun Yat-sen University. Prior to data collection, documented informed consent was acquired for every individual participant.

### 2.2. Clinical and Biochemical Measurements

A criterion-based questionnaire was administered to collect data on living style, health history, sociodemographic features, and histories of the family. Tobacco or alcohol consumption behaviors were grouped as “never,” “current” (regular smoking or drinking in the last 6 months), or “ever” (quit consuming tobacco or alcohol for over 6 months). The International Physical Activity Questionnaire (IPAQ) short table was adapted to count bodily activity during recreational periods, with questions added regarding how often and for how long moderate or vigorous activity and walking took place [13]. Total physical activity was assessed using individual metabolic equivalent hours per week (MET-h/week).

With the assistance of trained personnel, all candidates finished the anthropometric examinations based on a standard protocol. Duplicate blood pressure measurements were taken three times by a single observer, with a 5-min interval between each measurement. All data were acquired from an automatic electronic device (OMRON, Omron Corporation, Shanghai, China). Values averaged over the three blood pressure measurements were used in the ultimate data analysis. Participants were dressed without shoes and in light indoor clothes, and then height and weight were noted to the nearest 0.1 cm and 0.1 kg, separately. Body weight (kg) was divided by height (m) squared (kg/m^2^) to obtain body mass index (BMI) data. Next, subjects assumed a standing posture, and waist circumference was taken at the umbilical levels at the termination of a soft exhalation.

Following a one-night fast of a minimum of 10 h, a venous blood sample was obtained for bench testing. Fasting serum insulin, fasting plasma glucose (FPG), oral glucose tolerance test (OGTT) 2-h glucose, γ-gamma-glutamyl transpeptidase (γ-GGT), TG, total cholesterol (TC), HDL-C, LDL-C and creatinine were measured by an automated analyzer (Beckman CX-7 Biochemical Autoanalyser, Brea, CA, USA). The level of non-HDL-C was derived from the gap between serum TC and HDL-C. Hemoglobin A1c values (HbA1c) were available by high-performance liquid chromatography (Bio-Rad, Hercules, CA, USA). eGFR was computed in terms of ml/min per 1.73 m^2^ using the Modification of Diet in Renal Disease equation (MDRD) equation, expressed as eGFR = 186 × [serum creatinine × 0.011] − 1.154 × [age] − 0.203 × [0.742 if female] × 1.233 [14]. The diagnosis of diabetes was based on the 1999 World Health Organization diagnostic criteria.

### 2.3. Definition of CKD

The UACR is determined through dividing the albumin level by the creatinine level of the urine and denoting it as mg/g. An elevated amount of urine albumin was described as an UACR higher than or equivalent to 30 mg/g. CKD was defined by an eGFR below 60 mL/min/1.73 m^2^ or the development of albuminuria (UACR higher than or equivalent to 30 mg/g).

### 2.4. Statistical Analysis

Data were statistically analyzed with SAS version 9.2 (SAS Institute Inc., Cary, NC, USA). With biased variables, data are shown in the median (quartile range). Categorical variants are represented in figures or scales. Sequential variants are indicated by the mean ± standard deviation (SD). Intergroup differences were examined by one-way ANOVA. The χ^2^ test was applied for comparisons between categorical variables.

A chi-square test was used to identify the occurrence of CKD across different quartiles of baseline lipid parameters. Pearson’s correlation and multiple regression analysis models were applied to estimate the correlations of baseline lipid profiles with UACR, creatinine and eGFR at follow-up. Multiple Cox regressions were performed to assess the relationship of baseline lipid parameters and CKD risk. Hazard ratios (HRs) and corresponding 95% CIs for CKD were obtained according to each of the three models. In the three models, the first was unadjusted, the second was age-adapted, and the third was additionally adapted for sex, BMI, current smoking status, current drinking status, physical activity level and previously diagnosed dyslipidemia. Meanwhile, the area under the receiver operating characteristic (ROC) curve (AUC) was calculated to evaluate lipid parameters for diagnostic purposes. Owing to the nonnormal distribution of UACR and TG, they were subjected to log transformation prior to statistical analysis. The cigarette use condition and the alcohol use condition (noncurrent/current) were to be treated as categorical variants. Subgroup analysis was performed to estimate the risk of CKD per increasing interquartile in TG/HDL-C among subgroups at follow-up. The model was adjusted with the same factors as in model III. Among them, age was stratified at 58 years, sex was divided into male and female, BMI was divided into normal, overweight and obese sub-strata, and central obesity, diabetes and hypertension were stratified by presence and absence. In interaction studies, we investigated feasible correlates that might change the association between the risk of incident CKD and lipid parameters separately. Measurements for interaction were undertaken by adding every stratum parameter, lipid parameter interquartile and the corresponding interaction terms (stratification parameter multiplied by lipid parameter interquartile) to the ultimate model.

All statistical tests were two-sided, and *p* values < 0.05 were regarded as statistically meaningful.

## 3. Results

### 3.1. Clinical Features of the Research Group

The average follow-up age for the 5345 participants was 55.7 ± 7.1 years, and the occurrence of CKD was 6.4% (340), with a duration of follow-up (3.6 ± 0.7) years. Table 1 shows the features of the participants grouped by the existence of CKD. In terms of lipid parameters, patients with CKD had lower HDL-C and higher TG, non-HDL-C/HDL-C and TG/HDL-C than patients without CKD (all *p* < 0.0001).

### 3.2. Relationship between Lipid Profiles and Clinical Factors Associated with Renal Function

In Table 2, Pearson correlation analysis indicated that parameters TG, HDL-C, non-HDL-C, non-HDL-C/HDL-C and TG/HDL-C were notably associated with UACR; TG, TC, HDL-C, non-HDL-C/HDL-C and TG/HDL-C were notably related to creatinine; and TG, HDL-C, non-HDL-C, non-HDL-C/HDL-C and TG/HDL-C were remarkably associated with eGFR (all *p* < 0.05). Through proceeding with multiple regression analysis and additional adjusting for age and sex, it was shown that TG, HDL-C, non-HDL-C/HDL-C and TG/HDL-C remained correlated with UACR, creatinine and eGFR, respectively (all *p* < 0. 0001).

### 3.3. Associations between Lipid Parameters and CKD

The incidence of CKD in different quartiles of lipid parameters is presented in Figure 2. CKD morbidity tended to rise with increasing TG, non-HDL-C/HDL-C and TG/HDL-C quartiles and declined with increasing HDL-C quartiles (*p* < 0.001 for all trends). In Table 3, Cox regression analysis was applied to predict the risk of CKD for every quartile increase in TG, TC, HDL-C, LDL-C, non-HDL-C, non-HDL-C/HDL-C and TG/HDL-C levels. In the Cox regression analysis, patients were considered at higher risk for developing CKD with changes in interquartiles of TG, HDL-C, non-HDL-C/HDL-C and TG/HDL-C. The connection between these lipid markers and CKD remained after further adjustment for other potential confounders in models two and three.

Across all Cox regression models, TG/HDL-C showed the highest association with CKD in comparison to other lipid parameters (Table 3). Additionally, among all AUC values, TG/HDL-C was highest (0.599, 95% CI: 0.568–0.630). To validate the robustness of this outcome, we carried out a stratified analysis to determine the HRs of CKD per quartile gain in TG/HDL-C in distinct subgroups. Such an association varied under the influence of different stratification factors, as shown in Figure 3. The link between TG/HDL-C and CKD incidence was meaningful in both age strata (≥58 and <58), in the female strata, in the BMI strata (normal and overweight), in both central obesity strata, in the subjects without diabetes stratum and in the subjects without hypertension stratum. The interaction term for the correlation between TG/HDL-C and CKD incidence by stratification factor between subgroups was not statistically significant.

## 4. Discussion

Determining early warning parameters for decreased glomerular filtration rate and urinary albumin excretion in kidney disease is of great importance. The results of the current study suggest that TG/HDL-C represents a better predictor of CKD than do other general lipid indicators and deserves more attention in clinical practice to mitigate risk for patients.

Metabolic disorder of lipid profiles may affect the development of CKD through various mechanisms. As lipids are insoluble or slightly soluble in water and are bound to proteins in plasma in the form of lipoproteins, dyslipidemia actually appears as dyslipoproteinemia [15]. Dyslipoproteinemia can lead to the accumulation of renal fat and impaired kidney function. Hypertriglyceridemia and related insulin resistance can promote increased lipid uptake by the kidney, directly causing pathological changes in renal constituent cells and promoting fibrosis in the glomerulus, which in turn accelerates the development of CKD [16,17,18]. Lipid accumulation in podocytes would also have a greater impact on renal function. Due to increased synthesis and decreased excretion, cholesterol gradually accumulates in the podocytes, eventually destroying the glomerular filtration barrier, affecting the filtration function of the kidney, and progressively favoring development of nephropathy with proteinuria [18,19,20]. On the other hand, the widespread inflammatory and stress response caused by hyperlipidemia also impairs intrinsic cells, which impairs renal function as well [16,21,22,23]. Lipoprotein functions are altered in patients with CKD, aggravating the consequences of lipid disorders. Recent laboratory work has found a significant decline in superoxide dismutase and a noticeable rise in malondialdehyde concentrations in patients with CKD due to dyslipidemia and oxidative stress [24].

Epidemiological evidence indicates that there is a close association of dyslipidemia with renal damage and related diseases. Previous study has shown that the non-HDL-C/HDL-C ratio is more sensitive in identifying individuals who are at high risk for CKD among Asian populations [25]. Nevertheless, recently, a cross-sectional study pointed out that TG/HDL-C is a stronger predictor of changes and progression among indicators related to nephrogenic function than are TG, non-HDL-C/HDL-C and various routine lipid parameters [15], which is consistent with the results obtained in the current study. Actually, in patients with CKD, dyslipidemia can also predict cardiovascular disease and mortality at an early stage [26]. Research suggests that future testing for lipoproteins other than LDL-C (e.g., TG-rich lipoproteins) is needed for early detection and reduction of atherosclerotic cardiovascular disease (ASCVD) risk in patients with CKD [27,28]. Considering that ASCVD and CKD inherently share the same pathogenesis and common risk factors, we speculated that lipid abnormalities may play a synergistic role in the development of the above diseases.

In the prediction of CKD, TG/HDL-C level is probably preferable to other lipid indicators for several reasons. Firstly, decreased lipase and downregulation of lipoprotein receptors in patients with dyslipidemia and renal impairment affect the normal metabolism of very low density lipoprotein (VLDL) and chylomicrons, ultimately leading to an accumulation of TG and an increased risk of atherosclerotic complications [29]. Moreover, results of long-term cohort studies showed that higher serum TG levels increased the risk of severe decline in renal function, even in healthy individuals [30,31]. Based on the above evidence, TG is not only an important indicator for patients with advanced CKD, but also plays a role in predicting early CKD. Secondly, HDL-C levels tend to decrease and acquire dysfunctional characteristics in patients with CKD. HDL-C functions mainly to reverse the transport of cholesterol in the body, and it has an antioxidant action [29,32]. Intravascular foam cell formation and atherosclerotic plaque formation increase when HDL-C decreases [2]. As the main component of HDL-C, apolipoprotein A1 has a protective effect against atherosclerosis and inflammation [33]. In consequence, HDL-C could decrease lipid deposition in the arterial wall and promote cholesterol clearance from the circulation, which might delay the occurrence of kidney dysfunction [2]. Thirdly, TG/HDL-C ratio seems to be better correlated with the severity of albuminuria and the occurrence of CKD than is a single measurement of either TG or HDL-C. In addition to lipidic toxicity, other mechanisms such as extracellular matrix deposition and oxidative stress injury may participate in related pathophysiological process of CKD [34]. As a result, it remains to be further studied whether TG/HDL-C levels can be associated with other metabolic abnormalities involved in renal damage and indirectly contribute to renal progression.

This study has some limitations. Firstly, several studies have revealed that higher apolipoprotein B are correlated with a greater risk of atherosclerotic events in CKD and may be associated with accelerated progression of CKD in diabetic patients [35,36]. Therefore, apolipoproteins should be taken into account when analyzing possible risk factors associated with proteinuria and renal damage. Secondly, we measured urinary albumin excretion from morning urine samples. In previous literature, it has been concluded that 24-h urinary protein quantity provide more consistent albumin excretion results [37]. Nonetheless, results of a point-in-time urine sample are also representative of the results of 24-h or repeated urine samplings [38]. Using a point-in-time sample to obtain an assessment of UACR is an economical, convenient and reliable alternative to conducting large epidemiologic specimen collections. Finally, previous studies have proposed a number of formulas to calculate GFR in different populations. We chose the MDRD formula, based on previous relevant national studies [39]. The intrinsic differences should be noted when other methods are used to calculate GFR in further studies.

## 5. Conclusions

In conclusion, we have reported that TG/HDL-C ratio was independently associated with a higher risk of CKD in the Chinese population than were other lipid parameters tested in the study. These results suggest that control of dyslipidemia would play a crucial role in the prevention of renal damage in middle-aged and elderly subjects.

## Figures and Tables

**Figure 1 nutrients-15-00112-f001:**
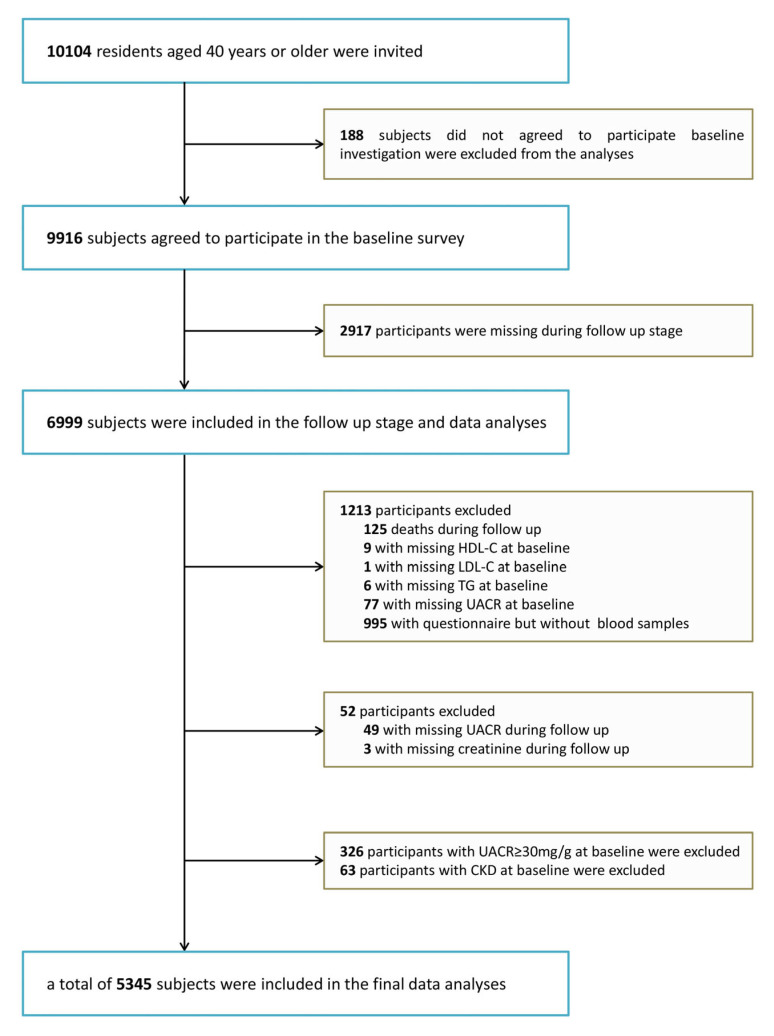
Flowchart of the population selection of the study.HDL-C, high-density lipoprotein cholesterol; LDL-C, low-density lipoprotein cholesterol; TG, triglycerides; UACR, urinary albumin-to-creatinine ratio; CKD, chronic kidney disease.

**Figure 2 nutrients-15-00112-f002:**
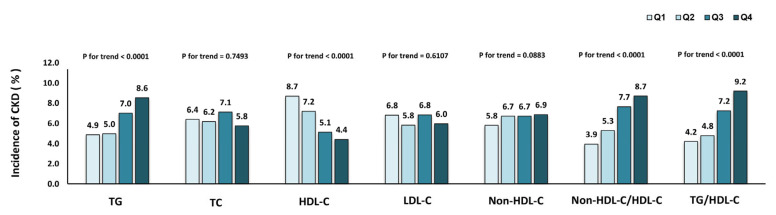
Incidence of chronic kidney disease in different quartiles of baseline lipid parameters.

**Figure 3 nutrients-15-00112-f003:**
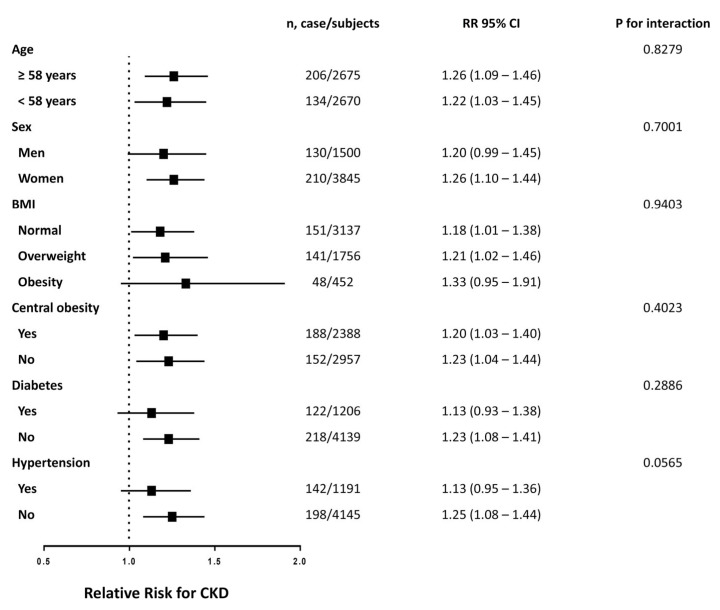
Risk of incident CKD with each quartile increase of TG/HDL-C in different subgroups at follow up. The model is adjusted for age, sex, BMI, current smoking status, current drinking status, physical activity level and previously-diagnosed dyslipidemia.

**Table 1 nutrients-15-00112-t001:** Baseline characteristics of study population by CKD status at follow up.

	without CKD	with CKD	*p*
n (%) *	5005 (93.6)	340 (6.4)	<0.0001
UACR (mg/g)	7.63 (5.59–10.79)	11.19 (7.15–16.79)	<0.0001
eGFR (ml/min per 1.73 m^2^)	103.1 ± 21.3	94.6 ± 24.5	<0.0001
TG (mmol/L)	1.25 (0.92–1.81)	1.49 (1.02–2.16)	<0.0001
TC (mmol/L)	5.22 ± 1.23	5.19 ± 1.24	0.6318
HDL-C (mmol/L)	1.34 ± 0.36	1.23 ± 0.33	<0.0001
LDL-C (mmol/L)	3.16 ± 0.95	3.13 ± 0.95	0.5898
Non-HDL-C (mmol/L)	3.89 ± 1.09	3.96 ± 1.09	0.2166
Non-HDL-C/HDL-C	3.05 ± 1.02	3.37 ± 1.05	<0.0001
TG/HDL-C	2.19 (1.47–3.48)	2.78 (1.85–4.69)	<0.0001
Age (years)	55.5 ± 7.0	58.5 ± 8.5	<0.0001
Male [n (%)]	1370 (27.4)	130 (38.2)	<0.0001
BMI (kg/m^2^)	23.5 ± 3.2	24.6 ± 3.3	<0.0001
WC (cm)	81.1 ± 9.3	84.2 ± 9.3	<0.0001
SBP (mmHg)	124.8 ± 15.6	132.5 ± 16.6	<0.0001
DBP (mmHg)	74.9 ± 9.7	77.5 ± 9.9	<0.0001
Current smoking [n (%)]	411 (8.4)	40 (12.1)	0.0215
Current drinking [n (%)]	156 (3.2)	13 (3.9)	0.4669
FPG (mmol/L)	5.40 (5.00–5.90)	5.60 (5.10–6.17)	<0.0001
OGTT 2 h glucose (mmol/L)	7.27 (6.09–9.00)	7.96 (6.53–10.29)	<0.0001
HbA1c	5.90 (5.60–6.20)	6.00 (5.70–6.40)	<0.0001
Fasting insulin (μIU/mL)	7.10 (5.20–9.60)	7.95 (5.85–11.20)	<0.0001
γ-GGT (U/L)	19.0 (14.0–28.0)	22.0 (15.0–30.0)	0.0047
Physical activity (MET-h/week)	22.0 (10.5–46.0)	24.5 (10.5–47.0)	0.9665

Data were means ± SD or medians (interquartile ranges) for skewed variables or numbers (proportions) for categorical variables. * n (%) was for the number of incident CKD status at follow up. *p* values were for the ANOVA or χ^2^ analyses across the groups. CKD, chronic kidney disease; UACR, Urinary albumin-to-creatinine ratio; eGFR, estimated glomerular filtration rate; TG, triglycerides; TC, total cholesterol; HDL-C, high-density lipoprotein cholesterol; LDL-C, low-density lipoprotein cholesterol; BMI, body mass index; WC, waist circumference; SBP, systolic blood pressure; DBP, diastolic blood pressure; FPG, fasting plasma glucose; OGTT, oral glucose tolerance test; γ-GGT, γ-glutamyltransferase; MET-h/week, separate metabolic equivalent hours per week.

**Table 2 nutrients-15-00112-t002:** Pearson’s correlation and multiple regression analysis of baseline lipid parameters associated with UACR, creatinine and eGFR at follow up.

	UACR (mg/g)				Creatinine (μmol/L)				eGFR (mL/min per 1.73 m^2^)			
	r	*p*	St. β	*p*	r	*p*	St. β	*p*	r	*p*	St. β	*p*
TG (mmol/L)	0.090	<0.0001	0.091	<0.0001	0.122	<0.0001	0.055	<0.0001	−0.107	<0.0001	−0.070	<0.0001
TC (mmol/L)	0.020	0.1486	−0.001	0.9231	−0.035	0.0100	−0.002	0.8720	−0.018	0.1866	0.001	0.9278
HDL-C (mmol/L)	−0.033	0.0163	−0.062	<0.0001	−0.193	<0.0001	−0.057	<0.0001	0.099	<0.0001	0.071	<0.0001
LDL-C (mmol/L)	0.006	0.6737	−0.011	0.4239	−0.023	0.0977	−0.008	0.4226	−0.012	0.3702	0.010	0.4583
Non-HDL-C (mmol/L)	0.034	0.0128	0.018	0.1899	0.021	0.1248	0.015	0.1539	−0.053	0.0001	−0.020	0.1319
Non-HDL-C/HDL-C	0.062	<0.0001	0.073	<0.0001	0.196	<0.0001	0.066	<0.0001	−0.140	<0.0001	−0.083	<0.0001
TG/HDL-C	0.089	<0.0001	0.104	<0.0001	0.187	<0.0001	0.071	<0.0001	−0.132	<0.0001	−0.090	<0.0001

UACR, Urinary albumin-to-creatinine ratio; eGFR, estimated glomerular filtration rate; TG, triglycerides; TC, total cholesterol; HDL-C, high-density lipoprotein cholesterol; LDL-C, low-density lipoprotein cholesterol. All parameters were logarithmically transformed prior to analysis due to non-normal distributions. r, correlation coefficient; St. β, Standardized regression coefficient; Multiple regression analysis is adjusted for age and sex.

**Table 3 nutrients-15-00112-t003:** Association between baseline lipid parameters and risk of CKD.

		Quartile 1	Quartile 2	Quartile 3	Quartile 4	1-Quartile Change #	AUC (95% CI) *
TG	Model 1	1	1.02 (0.72–1.45)	1.48 (1.06–2.05)	1.83 (1.33–2.50)	1.25 (1.13–1.38)	0.577 (0.545–0.609)
	Model 2	1	0.98 (0.69–1.39)	1.37 (0.99–1.91)	1.68 (1.22–2.31)	1.22 (1.10–1.35)	
	Model 3	1	0.99 (0.69–1.42)	1.31 (0.93–1.85)	1.48 (1.06–2.08)	1.16 (1.04–1.29)	
TC	Model 1	1	0.96 (0.70–1.32)	1.12 (0.83–1.52)	0.89 (0.65–1.23)	0.98 (0.89–1.09)	0.495 (0.464–0.527)
	Model 2	1	0.94 (0.69–1.29)	1.08 (0.80–1.47)	0.81 (0.59–1.12)	0.96 (0.87–1.06)	
	Model 3	1	0.96 (0.70–1.33)	1.09 (0.80–1.50)	0.84 (0.60–1.17)	0.96 (0.87–1.07)	
HDL-C	Model 1	1	1.23 (0.93–1.63)	1.76 (1.29–2.40)	2.07 (1.50–2.85)	1.29 (1.17–1.43)	0.411 (0.381–0.441)
	Model 2	1	1.24 (0.94–1.65)	1.75 (1.28–2.38)	1.99 (1.44 - 2.75)	1.28 (1.15–1.41)	
	Model 3	1	1.19 (0.88–1.59)	1.53 (1.11–2.12)	1.62 (1.15–2.30)	1.19 (1.07–1.33)	
LDL-C	Model 1	1	0.85 (0.62–1.16)	1.00 (0.74–1.35)	0.87 (0.63 0 1.18)	0.98 (0.88–1.08)	0.495 (0.463–0.527)
	Model 2	1	0.83 (0.61–1.14)	0.96 (0.71–1.30)	0.78 (0.57–1.07)	0.94 (0.85–1.04)	
	Model 3	1	0.85 (0.62–1.18)	0.97 (0.71–1.33)	0.78 (0.56–1.08)	0.94 (0.85–1.04)	
Non-HDL-C	Model 1	1	1.32 (0.96–1.82)	1.32 (0.95–1.82)	1.35 (0.98–1.86)	1.09 (0.99–1.20)	0.526 (0.494–0.557)
	Model 2	1	1.27 (0.92–1.75)	1.23 (0.89–1.71)	1.21 (0.87–1.67)	1.05 (0.95–1.16)	
	Model 3	1	1.32 (0.95–1.85)	1.20 (0.86–1.68)	1.18 (0.84–1.65)	1.03 (0.93–1.15)	
Non-HDL-C/HDL-C	Model 1	1	1.36 (0.95–1.95)	2.02 (1.42–2.86)	2.32 (1.65–3.28)	1.33 (1.20–1.47)	0.595 (0.564–0.625)
	Model 2	1	1.33 (0.93–1.92)	1.88 (1.32–2.66)	2.07 (1.47–2.93)	1.28 (1.15–1.42)	
	Model 3	1	1.24 (0.85–1.81)	1.66 (1.16–2.41)	1.66 (1.15–2.42)	1.19 (1.06–1.33)	
TG/HDL-C	Model 1	1	1.15 (0.80–1.66)	1.78 (1.27–2.50)	2.31 (1.67–3.20)	1.35 (1.22–1.49)	0.599 (0.568–0.630)
	Model 2	1	1.08 (0.74–1.55)	1.66 (1.18–2.33)	2.11 (1.52–2.93)	1.32 (1.19–1.46)	
	Model 3	1	0.98 (0.67–1.44)	1.46 (1.03–2.10)	1.72 (1.22–2.46)	1.24 (1.11–1.38)	

Data are odds ratios (95% confidence interval). Participants without CKD at follow up are defined as 0 and with CKD as 1. # All variables were calculated for 1-Quartile increasing of lipid parameters except for HDL-C, which was calculated for 1-Quartile decreasing. * AUC (95% CI), area under the receiver operating characteristic curve (AUC) and the corresponding 95% confidence intervals (CI). Model one is unadjusted. Model two is adjusted for age. Model three is adjusted for age, sex, BMI, current smoking status, current drinking status, physical activity level and previously diagnosed dyslipidemia.

## Data Availability

Data are available upon request from the corresponding author.

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
