# Peer review of "Lipid Parameters and the Development of Chronic Kidney Disease: A Prospective Cohort Study in Middle-Aged and Elderly Chinese Individuals"

_nutrients, 2022, doi:10.3390/nu15010112_

Round 1
Reviewer 1 Report
The manuscript of Liao S. et al evaluated the association of conventional lipid markers with renal function in a prospective cohort study. The study was carried out in middle-aged and elderly Chinese individuals between 2011 and 2014 to 2016. The main finding of the study is that TG/HDL-C ratio would be the most effective predictor of Chronic kidney disease, at least in the studied population. Although the the work could provide interesting data regarding the association between lipid metabolism disorders and the progression of chronic kidney disease, this reviewer considers that some major revisions should be made:
Major revisions
1- In section 2.3 Definition of CKD the authors stated that: CKD was defined by an eGFR below 60 ml/min/1.73 m2 or the development of albuminuria (ACR higher than or equivalent to 30 mg/g). However, the participants included in the CKD group does not meet these criteria, since the values showed in Table 1 for eGFR are higher than 60 ml/min/1.73 m2 and the values for UACR are less than 30 mg/g. How do the authors explain the inclusion of these participants in this group? It is very important since a condition for the study to be relevant is that there is a population with chronic kidney disease.
2- The comparison of the parameters studied, shown in Table 2, is made between two very different populations, one of them has n=5005 and the other has n=340. Could the imbalance in the number of individuals that conform each group influence the statistical analysis of the results obtained?
3- Table 2 shows that the relation between non-HDL-C and creatinine have a P value of 0.1248. This means that the correlation is not statistically significant. However, in the section 3.2. Relationship between lipid profiles and clinical factors associated with renal function the authors mention that ….non-HDL-C….. were notably related to SCr. If the P value shown in the table is correct then this statement would not be correct. On the other hand, regarding to the values obtained in the correlations: the values of Pearson's r are mostly very close to 0 (zero), which suggests a low correlation. Can the authors then affirm that there is a significant correlation between the parameters studied?
Minnor comments
1- The objective of the work is described in the introduction section but not in the abstract. It should be included in the abstract.
2- Please include the definition of the abbreviations used the first time they appear. P. 1 line 38: Please include the meaning of the abbreviation eGFR , P. 2 line 60: Please include the meaning of the abbreviation UACR, P.4 line 134: Please include the meaning of the abbreviation SCr.
3-P. 3 line 103: The authors use the term albumin-to-creatinine ratio and the abbreviation ACR (this abbreviation was also used in section 2.2 Clinical and biochemical measurements), but the term UACR is used in the introduction, in p4. Line 155 and in the Results section. The same abbreviation should be used along the manuscript.
4-P.3 line 96: The authors stated that “In all, 10,104 participants were enrolled who were 48 years or older…”. However, Figure 1 shows that “ 10 104 residents aged 40 years or older were invited. Please correct the age of the participants.
5-The quality of Figures should be improved
6- Table 1: Serum lipid levels, including total cholesterol levels (TC) are shown in mmol/L. However, non-HDL-C levels are shown in mg/dL. Since the levels of non-HDL-C was derived from the gap between serum TC and HDL-C, it would be more appropriate to express non-HDL-C in mmol/L.
7- Table 1 shows the serum levels of γ-GGT (U/L). However, section 2.2 Clinical and biochemical measurements does not include this determination. The determination of γ-glutamyltransferase must be included.
8- In 3.2 Results section the authors use the term SCr instead of creatinine (as shown the Table 2). It would be more appropriated to use the same term SCr or creatinine.
9- Serum lipid levels were shown in mmol/l in table 1 and mg/dL in table 2. It would be more appropriate to express them in the same units throughout the manuscript.
10-P.10 lines 299-304: The inclusion of apo M in the discussion is not clear. This sentence should be rewritten indicating more clearly the role of apo B, which explains its need to be studied in the future, and separately the role of the apo M.
11- The Bibliography is up-to-date, however it must be checked and corrected since some of them are incomplete (eg: volume, page numbers.)
Reviewer 2 Report
The manuscript titled "Lipid parameters and the development of chronic kidney disease" by Shumei Liao and others is a fairly good manuscript. Apart from minor changes, I consider the work to be published.
Below are my comments and suggestions.
1. I don't know why inserts like "Background" and others in the abstract. After all, when reading a work, you know where it is and what it is about. I recommend removing this.
2. The list of keywords should be completed. The manuscript is probably a bit broader thematically.
3. Besides, I don't know if so many figures in the abstract are really necessary.
4. It seems to me that the introduction is too poor in relation to key words. It definitely needs to be replenished. Even after entering the keywords provided by the authors of the work, I can see that it is quite an interesting and important topic.
5. In the last paragraph of the introduction, please clearly separate the purpose of this manuscript.
6. Fig. 1 terrible quality. Please correct.
7. Fig 2 - also to be improved.
8. Fig 3 - it's a shame to even discuss it. Tragic quality. I am surprised at all by the authors that they send the manuscript to a journal with such a high IF and they are not ashamed of the poor quality of figs. After all, by saving the work as a pdf, they probably checked what they came up with.
It really surprises me.
9. I can't see anything but one poor paragraph in the summary section. Please do it more carefully.
To sum up, the work seems really valuable and well thought out, but its shortcomings are glaring.
I recommend a thorough revision.
Round 2
Reviewer 1 Report
-In the revised manuscript the authors shown non-HDL-C serum levels in mmol/L (as the reviewer suggested), however the values were not corrected (the values are in mg/dL) e.g: the group without CKD have a value of TC of 5.22 ± 1.23mmol/L, HDL-C 1.34 ± 0.36 mmol/L and non-HDL-C of 150.1 ± 42.1 (this value is not in mmol/L) please correct.
Reviewer 2 Report
The authors answered my questions and suggestions very well. I suggest accepting the manuscript.
Author Response
Response to Reviewer 2 Comments
Comments and Suggestions for Authors
-The authors answered my questions and suggestions very well. I suggest accepting the manuscript.
Response:
Thank you for your revision again and we appreciate very much for all of the previous comments. The manuscript has certainly benefited from such insightful suggestions.